# Efficient Large-Scale and Scarless Genome Engineering Enables the Construction and Screening of *Bacillus subtilis* Biofuel Overproducers

**DOI:** 10.3390/ijms23094853

**Published:** 2022-04-27

**Authors:** Jiheng Tian, Baowen Xing, Mengyuan Li, Changgeng Xu, Yi-Xin Huo, Shuyuan Guo

**Affiliations:** Key Laboratory of Molecular Medicine and Biotherapy, Beijing Institute of Technology, School of Life Science, No. 5 South Zhongguancun Street, Beijing 100081, China; 18801003306@163.com (J.T.); xingbwsd@163.com (B.X.); mengyuaninuf@outlook.com (M.L.); xuchanggeng66@163.com (C.X.)

**Keywords:** CRISPR-Cas9, *Bacillus subtilis*, genome engineering, long fragment deletion, metabolic engineering

## Abstract

*Bacillus subtilis* is a versatile microbial cell factory that can produce valuable proteins and value-added chemicals. Long fragment editing techniques are of great importance for accelerating bacterial genome engineering to obtain desirable and genetically stable host strains. Herein, we develop an efficient CRISPR-Cas9 method for large-scale and scarless genome engineering in the *Bacillus subtilis* genome, which can delete up to 134.3 kb DNA fragments, 3.5 times as long as the previous report, with a positivity rate of 100%. The effects of using a heterologous NHEJ system, linear donor DNA, and various donor DNA length on the engineering efficiencies were also investigated. The CRISPR-Cas9 method was then utilized for *Bacillus subtilis* genome simplification and construction of a series of individual and cumulative deletion mutants, which are further screened for overproducer of isobutanol, a new generation biofuel. These results suggest that the method is a powerful genome engineering tool for constructing and screening engineered host strains with enhanced capabilities, highlighting the potential for synthetic biology and metabolic engineering.

## 1. Introduction

Synthetic biology has made rapid development, showing great potential in biosensing, therapeutics and in the production of novel biomaterials [1,2]. In synthetic biology, the expression of genetic components requires a good chassis as a basis [3,4,5]. More and more “cell chassis” with simplified genomes and metabolic networks are being developed [6]. Using a minimal cell as a production chassis can reduce transcription costs, eliminate competing pathways, and limit unwanted regulatory pathways [7]. Both “top-down” and “bottom-up” approaches have been used in constructing a minimal cell [8,9,10]. The “bottom-up” approach requires that the synthesized pathways be assembled into long fragments from scratch and that all essential components be ligated into the chromosome [7,11,12,13,14]. However, with “bottom-up” approaches it is difficult to maintain genomic stability because of the complexity of metabolic processes and interactions in the organism [12,15]. The “top-down” approach refers to building a cell factory by deleting nonessential genes and metabolic pathways from the natural cell [10,16,17,18], which is a more pragmatic approach than “bottom-up” for building an ideal chassis [7,15].

Currently, global environmental and resource issues need to be addressed urgently. Higher alcohols such as isobutanol, n-butanol and 3-methyl-1-butanol (3MB), etc., are seen as an ideal alternative to gasoline due to their high energy density and low hygroscopicity [19,20,21]. To be able to obtain high yields and titers of products, microorganisms used for biofuel production are required to be able to tolerate the stresses in the process [22]. *Bacillus*
*subtilis* (*B. subtilis*), a Gram-positive GRAS (generally recognized as safe) bacterium [23,24,25], which is highly tolerant of high concentrations of isobutanol, is considered to be an excellent industrial host bacterium for the production of bulk chemicals [26,27] such as isobutanol [28,29]. Taken together, the construction of *B. subtilis* with high isobutanol production is of great importance.

To minimize the risk of the disruptive effects of the production of bulk chemicals on the cells themselves and on metabolites [30,31], the *B. subtilis* genome could be simplified to eliminate nonessential features, concentrate cellular metabolic activity on the desired products and reduce the cost of industrial production [32,33]. Therefore, the ideal chassis should contain only the minimum set of functions required for synthetic production [12,34]. Long fragment editing technology was used to accelerate the modification of bacterial genomics and obtain an ideal chassis to construct ideal hosts for producing various drugs, proteins, and small-molecule chemicals, etc. [35].

Westers et al. constructed a mutant strain with a 7.7% reduction in the genome, without changing the growth rate [36]. Both Ara et al. [37] and Kumpfmuller et al. [38] improved the yield of industrial production by simplifying the genome. Tanaka et al. successfully deleted 146 intervals ranging in length from 2.0 to 159.0 kb, covering 76.0% of the chromosomes [39,40]. Morimoto et al. sequentially deleted up to 20.7% of the DNA sequence (874.0 kb), while the production of extracellular cellulase and protease significantly increased [41]. To improve the production of guanosine and thymine, Li et al. sequentially deleted 814.4 kb of nonessential domains in the *B. subtilis* genome [42]. The traditional genome editing methods in *B. subtilis* have achieved good results. Nevertheless, it should not be overlooked that selection markers limit multiple modifications of the genome [43,44] and that recombinase plasmids require secondary transformation. Most importantly, these traditional methods are inefficient and scar the chromosome [26,45].

The CRISPR-Cas (clustered regularly interspaced short palindromic repeats-CRISPR-associated protein) system (Appendix A) that defends against an invasion by phages and plasmids is a class of immune system proteins widely found in bacteria and archaea [46,47]. In this system, crRNA and tracrRNA are fused into a single-stranded guide RNA (sgRNA) [48]. The 20-nucleotide (20-nt) sequence at the 5′ end of the sgRNA is complementary to the target site. The 3′ end of sgRNA is connected to the Cas9 protein through a linker loop, which does not affect the spatial structure [49]. Under the guidance of the sgRNA, the Cas9 protein cleaves DNA, thereby causing the double-stranded break (DSB) [50] at the target site where a proper protospacer-adjacent motif (PAM) exists, which is 5′-NGG-3′ for *S. pyogenes* Cas9 [25]. The DSB can be repaired by error-prone nonhomologous end-joining (NHEJ) or precise homology-directed repair (HDR) [49,51]. In *B. subtilis*, there is a natural NHEJ system, which is mainly active in the late growth and spore formation phase [25,52]. Moreover, *B. subtilis* does not require the assistance of a heterologous homologous recombination system, and the exogenous DNA becomes single-stranded and efficiently recombines with homologous chromosomes as it enters the cell [53,54]. The CRISPR-Cas9 gene-editing technology breaks through the limitations of traditional methods, improves editing efficiency, and accelerates genome simplification [55]. Altenbuhner proposed a single plasmid system to introduce a 25.1 kb long fragment deletion in the *B. subtilis* genome [53]. So et al. developed a dual plasmid system in which one plasmid carried the Cas9-encoding gene and the other carried the sgRNA and donor DNA. This system introduced 38.0 kb of PPS operon deletion, with a positivity rate of 80% [56]. Song et al. used the CRISPR-Cas9 system to integrate a 2.5 kb expression cassette into the *B. subtilis* genome [57]. There is no doubt that CRISPR-Cas9 gene-editing technology is feasible and has great potential in *B. subtilis*. However, the existing CRISPR-Cas9 system is not mature enough for the application of genome engineering.

Large genomic deletions by CRISPR-Cas9 have rarely been reported in *B. subtilis*. To further advance the construction of the minimal genome of *B. subtilis* and obtain improved chassis cells, we optimized the existing CRISPR-Cas9 system for long fragment editing with high efficiency. The optimized system was able to achieve long fragment deletion for fragments of up to 134.3 kb, simultaneous two fragment deletions, and the insertion of the purple protein operon. The linear donor DNA and NHEJ system were also applied successfully for genome editing in *B. subtilis*. Additionally, a series of genome-simplified strains were explored in metabolic engineering to construct the isobutanol overproducers. Eventually, an overproducer with an isobutanol titer 2.4 times the titer produced by the wild-type strain was successfully obtained.

## 2. Results

### 2.1. NHEJ Is Ineffective in B. subtilis for Long Fragment Deletion

According to previous reports, the wild-type *B. subtilis* 168 chromosome is 4.21 Mb and includes more than 4100 protein-coding genes (CDS), accounting for 87% of the entire genome sequence [58,59]. According to recent studies, there are 257 essential genes for bacterial growth in LB medium at 37 °C, most of which are related to cell membrane structure, genetic information transfer, and energy metabolism [58,60,61]. This provides us with important information for simplifying nonessential long fragments of the genome.

Although homologous recombination-mediated genome engineering is efficient and accurate, it is limited by the complexity of the process of constructing DNA editing templates [62]. Unlike most prokaryotes, a highly conserved NHEJ system also exists in eukaryotic cells to maintain genomic stability (Appendix A) [63,64]. In previous reports, Su et al. [62] and Zheng et al. [65] introduced a *Mycobacteria*-derived NHEJ pathway into *E. coli* to repair the DSBs generated by CRISPR-Cas9. Although a similar system has been shown to exist in *B. subtilis* NHEJ [63], it is inefficient [65]. In contrast, the NHEJ pathway derived from *Mycobacterium tuberculosis* (Mtb) H37Rv exhibits good efficiency [62,65]. Therefore, in combination with the CRISPR-Cas9 system developed by Altenbuchner [53], we constructed a two-plasmid system in which the CRISPR system and the NHEJ system were expressed separately. One plasmid, pCas-sgRNA-X, carrying the CRISPR-Cas9 system, expressed Cas9 to cleave at specific sites on the DNA double strands. Another plasmid, pHT01-Mtb-NHEJ, carries the Mtb-NHEJ system in *B. subtilis* by overexpressing the adjacent *ligD* and *mku* genes driven by the IPTG-inducible P_grac_ promoter. The DNA end-binding protein Mtb-Ku is responsible for recognizing and stabilizing DSBs, while Mtb-LigD is recruited to DNA ends for repair, resulting in insertion and/or deletion (indel) mutations (Figure 1a).

To verify the efficiency of the NHEJ system, we managed to inactivate the extracellular proteases genes *nprB* and *vpr* in *B. subtilis*, which achieved satisfactory results. Theoretically, the repair process would be accompanied by random deletions of DNA sequences, which disrupt the open reading frames (ORFs) of *nprB* and *vpr*. After the addition of IPTG and mannose to induce editing, four single colonies randomly picked from the *nprB* gene editing plate were verified by PCR using primers *nprB*-F/R (Figure 1a). In principle, if there was a deletion in the *nprB* gene sequences, the PCR product would be significantly smaller than the culture. As a result, the band in lane 4 was significantly smaller than those in the other lanes including lanes 1–3 and the control (Figure 1b). As demonstrated by Sanger sequencing, 496 bp was successfully deleted in the *nprB* gene with a positivity rate of 25% (Figure 1b). The plasmid pCas-sgRNA-*nprB* was cured by increasing the culture temperature to 50 °C, and a new plasmid pCas-sgRNA-*vpr* targeting a different genomic locus was transformed for the next round of gene editing. Similarly, the verification of *vpr* gene editing by primers *vpr*-F/R (Figure 1a) showed that the band in lane 3 is significantly smaller than those in the other lanes. As demonstrated by the Sanger sequencing, 490 bp was successfully deleted in the *vpr* gene with a positivity rate of 25% (Figure 1b). Ultimately, using this system, a mutant strain GN2 was obtained that successfully inactivated the *nprB* and *vpr* genes.

Then, we tried to apply this effective system to delete long fragments in the *B. subtilis*. To delete a 17.6 kb fragment, we designed the plasmid pCas-sgRNA-17.6 kb, containing two sgRNA-expression cassettes, one targeting the *alkA* gene and the other targeting a nonsense sequence downstream of the *skfH* gene (Appendix A). Using primers F1/R1 and F2/R2, 23 colonies were randomly selected for PCR screening. As a result, the primers F1/R1 targeting outside the 17.6 kb fragment could not amplify any PCR product, while the primers F2/R2 targeting inside the deleted fragment all obtained the same band as the control. Unfortunately, it did not provide successful long fragment deletion in *B. subtilis* (Appendix A).

### 2.2. Long Fragment Deletion

Therefore, we switched back to the homologous recombination (HR) system, although the system was identified as inefficient and inconvenient for long fragment deletion in previous reports. We tested and optimized based on the system developed by Altenbucher [53]. To delete a long fragment, we first constructed one plasmid system containing an sgRNA-expression cassette, Cas9-encoding gene and donor DNA as described in previous reports. For each long fragment deletion, two pairs of primers for PCR verification were designed. The first primer pair targeted the adjacent sequences outside the two homologous arms (HAs) and the second primer pair targeted DNA sequences within the deleted area. In theory, the correctly edited clones would show the expected size of PCR products with the first primer pair and nothing would show up with the second. On the contrary, the unedited clones would show the corresponding PCR products with the second primer pair but nothing with the first.

First, we constructed the plasmid pCas-sgRNA-1 containing a sgRNA-targeting *alkA* gene within the 17.6 kb fragment (Appendix A). Then, 23 randomly selected colonies were screened using primers F1/R1, 19 of which successfully amplified the 1880 bp band (Appendix A). An 829 bp band of the same size as the control was obtained for unedited colonies using primers F2/R2. Thus, the 17.6 kb fragment was successfully deleted using the HR-mediated method with a positivity rate of 82.6%, which proved that this system was functional.

Next, we used the same method to design eight plasmids, containing different donor DNA and sgRNA-expression cassettes targeting 20.0, 25.9, 27.8, 32.9, 35.6, 38.1, 48.9 and 52.3 kb fragments, respectively. In a previous report, similar fragments were successfully deleted by Li et al. [42], which demonstrated that these fragments distributed in different locations of the chromosome were nonessential for *B. subtilis*. For each fragment deletion, we randomly verified 23 single colonies, respectively. For the colonies that successfully deleted the 25.9 kb fragment, a 1706 bp band was successfully amplified using primers F5/R5, and a 640 bp band consistent with the control could not be obtained using primers F6/R6. As shown in Appendix A, 22 single colonies were successfully edited, with a positivity rate of 95.7%. Similarly, only four colonies were successfully deleted for the 35.6 kb long fragment (Appendix A), with a positivity rate of 17.4% (Figure 2b), which seemed to indicate the instability of this method. For the remaining six long fragment deletions (20.0, 27.8, 32.9, 38.1, 48.9 and 52.3 kb), the results were unsatisfactory. Even though we changed the different target sites and extended the induction time, we did not screen for correctly edited colonies. This was most likely because the DSB was too far away from the N-terminal or C-terminal recombination regions to effectively initiate the double exchange efficiently. Moreover, the introduction of only one DSB into the genome was unstable and did not guarantee the complete deletion of the entire long fragment. Therefore, we optimized the system by targeting and cleaving two ends of the fragment in the genome.

For the 20.0 kb fragment deletion, we designed a plasmid pCas-sgRNA-2 containing two sgRNAs that targeted two nonsense sequences upstream of the *ynzG* and *yncE* genes, respectively. As shown in Appendix A, the primers F3/R3 and F4/R4 were used to verify the editing results, which showed that seven colonies successfully deleted the 20.0 kb fragment, with a positivity rate of 30.4%. Using the same approach, 27.8, 32.9, 38.1, 48.9, 52.3, 81.1 and 98.0 kb long fragments were successfully deleted, with positivity rates of 21.7%, 34.8%, 91.3% and 78.2%, respectively (Figure 2b). Notably, the largest 134.3 kb fragment was deleted with high efficiency by designing two sgRNAs, one targeting the *yotM* gene and the other targeting the *sprA* gene (Figure 3b), and 2000 bp donor DNA. All 23 randomly selected colonies were identified as deletion mutants by PCR verification and Sanger sequencing with a positivity rate of 100% (Figure 3c). In conclusion, we successfully deleted 12 nonessential fragments in the *B. subtilis* genome, ranging in length from 17.6 to 134.3 kb. The location of these long fragments in the genome is shown in yellow in Figure 2a, and the target sites and primer locations used are outlined in Appendix A. Compared to the two existing CRISPR-Cas9-based methods in *B. subtilis* [53,56], our method achieved better editing activity, with up to 134.3 kb of genomic DNA deletion and a positivity rate of 100%.

### 2.3. Effect of Donor DNA Size on Positivity Rate

Theoretically, the size of donor DNA affects the efficiency of HDR. However, the large size of donor DNA would increase the difficulty of plasmid construction and transformation. To explore the shortest donor DNA required for long fragment deletion, we truncated the length of the donor DNA from 2000 to 1500, 1000, 600 and 360 bp (Figure 4a). Based on plasmid pCas-sgRNA-12, we shortened the intermediate part of donor DNA and constructed plasmids pCas-12-1500, pCas-12-1000, pCas-12-600 and pCas-12-360, respectively (Figure 4a). We designed an overlap of 20 bp at the 5′ end of each primer pair and performed SOE-PCR with plasmid pCas-sgRNA-12 as the template. The obtained PCR products can be self-conjugated in *E. coli* JM109. All editing results can be verified using primers F23/R23 and F24/R24 (Figure 3b). If 2000, 1500, 1000, 600 and 360 bp donor DNA-mediated homologous recombination was successful, the edited colonies could be verified using primers F23/R23 to obtain PCR products of 2344, 1782, 1282, 882 and 642 bp, respectively. Each plasmid was transformed into wild-type *B. subtilis* and the edited colonies were obtained after induction by mannose. Previously, we verified that the 134.3 kb fragment can be deleted using plasmid pCas-sgRNA-12 with a positivity rate of 100% (Figure 3c). We randomly picked 10 single colonies from four editing plates of the 134.3 kb fragments to verify the editing efficiency of plasmids pCas-12-1500, pCas-12-1000, pCas-12-600 and pCas-12-360. As a result, successfully edited colonies were obtained using 1500 and 1000 bp donor DNA as editing templates with a 100% positivity rate (Figure 4b,c). Although using 600 and 360 bp of donor DNA as the editing template was also able to delete the 134.3 kb long fragment, the positivity rates were only 30% and 10%, respectively (Figure 4b,c). This suggested that when using homologous recombination to delete long fragments in the *B. subtilis* genome, the donor DNA should be longer than 600 bp to ensure high editing efficiency.

### 2.4. Achieving Long Fragment Deletions Using Linear Donor DNA

Although the efficient deletion of the 134.3 kb long fragment could be achieved, the construction of the corresponding plasmid containing two sgRNA-expression cassettes was inconvenient, which was unhelpful when simplifying the *B. subtilis* genome rapidly. Therefore, based on recombinant engineering assisted by the CRISPR-Cas9 system, we managed to use linear donor DNA as an editing template (Figure 5a), which is novel in *B. subtilis*. Previously, we verified that 81.1 and 134.3 kb long fragments can be successfully deleted using circular donor DNA as the editing template, with a positivity rate of 100% for both (Figure 3c and Appendix A). When using linear donor DNA as a repair template, to reduce the degradation of the linear donor DNA by DNA exonuclease during culture enrichment, we transformed the linear DNA template together with plasmid pCas-sgRNA-X. Then, the transformation mixture was directly painted on the plates with antibiotics and mannose. We randomly picked 23 single colonies on 81.1 and 134.3 kb editorial plates for verification, respectively. For the 81.1 kb fragment deletion, only lanes 1 and 22 yielded correct bands, indicating that these two colonies were successfully edited using linear donor DNA, with a positivity rate of 8.7% (Figure 5b,c). For the 134.3 kb fragment deletion, all 20 colonies were successfully edited with a positivity rate of 87.0%, except for lanes 10, 12 and 18, where erroneous bands were obtained (Figure 5b,c). Although the use of linear donor DNA for genome editing simplified the plasmid construction step, the sensitivity of the exonuclease to linear donor DNA could lead to reduced editing efficiency. In this respect, we can avoid this problem by increasing the concentration of donor DNA as much as possible. For example, in the present study, we ensured that the amount of the transformed linear donor DNA fragment was approximately 2 µg in the 500 µL transformation mixture.

### 2.5. Chromosomal Simplification

To establish the minimal genome, we began with constructing cumulative deletion mutants based on the strain Guo1. Through iterative editing, we obtained strain Guo13 with deletions of 17.6, 20.0, 38.1 and 98.0 kb fragments. However, multiple rounds of editing are time-consuming. To shorten the time required, we built one plasmid pCas-sgRNA-3/12 (Appendix A) comprising three sgRNA-expression cassettes to delete 25.9 and 134.3 kb long fragments at the same time. The sgRNA3 and the donor DNA from the plasmid pCas-sgRNA-3 were integrated into plasmid pCas-sgRNA-12 (Appendix A). The primers F5/R5 and F6/R6 were used to verify the 25.9 kb fragment deletion, and the F23/R23 and F24/R24 primers were used to verify the 134.3 kb fragment deletion. As shown in Appendix A, all 23 randomly picked single colonies successfully deleted the 134.3 kb fragment, and 19 of them deleted the 25.9 kb fragment at the same time. As a result, 82.6% of the randomly picked single colonies had both 25.9 and 134.3 kb fragment deletions (Appendix A). The mutant strain Guo14 (Δ17.6, 20.0, 25.9, 38.1, 98.0 and 134.3 kb), after knocking out a total of 334.0 kb DNA sequence, was therefore obtained (Figure 2a, regions in green color). This approach allowed multiple fragment deletions in just one round, which could greatly accelerate genome simplification in the future. In summary, our improved method is efficient and stable, facilitates the deletion of a greater number of long fragments, and the construction of minimal genomic cells is just around the corner.

We tried to verify the effect of long fragment deletion on the phenotype of the strain. We measured the growth curves of the 10 deletion mutants (Guo1, Guo2, Guo3, Guo5, Guo6, Guo7, Guo8, Guo10, Guo11 and Guo12). After about 8 h of growth, these strains reached a stable stage. There were no significant differences in growth rates between the deletion mutant strains and the wild-type strain (Appendix A), indicating that the deletion of these fragments did not affect cell growth. Using microscopy, it was observed that the deletion of these long fragments also did not change the morphology of the bacteria, except for the deletion of the mutant strain Guo11 (Δ98.0 kb) (Figure 2c). Almost all strain Guo11 in the field of view were identified as elongated filamentous with microscopic observation. An analysis of the 98.0 kb long fragment revealed that most of the genes in this region are membrane and transporter-protein-related genes, including the *yxaB* gene, which is associated with biofilm formation. It was probable that the deletion of these genes led to the inhibition of cell division and elongated bacterial morphology.

### 2.6. Gene Insertion

Besides long fragment deletion, this optimized system can also be used for gene insertion. We attempted to integrate the purple protein operon into the *B. subtilis* genome. Based on plasmid pCas-sgRNA-12, we designed and constructed plasmid pCas-12-purple. The purple protein operon, containing the P_grac_ promoter, was inserted into the middle of the donor DNA (Figure 6a). Thus, the editing template, LHA-P_grac_-purple-RHA, was able to integrate the purple protein operon into the genome during the repair process (Figure 6b). Three pairs of primers were designed to verify the insertion of the operon. Three single colonies were randomly picked and inoculated into the LB medium at 50 °C to cure the plasmid pCas-12-purple. Then, 134.3 kb fragment deletion was verified using primers F23/R23 and F24/R24, and primer PP-F/R was used to verify purple protein operon insertion. For a correctly edited single colony, the PCR product obtained using primers F23/R23 should have a length of 2869 bp, while the PCR using primers F24/R24 had no bands (Figure 6c). As shown in Figure 6c, all three colonies successfully deleted the 134.3 kb long fragment, but the bands of lanes 1 and 2 verified using primers F23/R23 appear to be lower than the bands of lane 3. Additionally, the results verified using primer PP-F/R showed that only lane 3 had the correct 1143 bp band (Figure 6c). To further confirm whether the purple protein was successfully inserted into the genome, we inoculated the three strains into an LB liquid medium without antibiotics and attempted to express the purple protein. Since the operon did not have the *lacI* gene, the promoter P_grac_ became a constitutive promoter and no longer required IPTG induction. After induction at 30 °C for 24 h, the bacterial fluid was collected and the supernatant was poured off by centrifugation. It was found that the precipitate in EP tube No. 3 showed a lavender color, while the precipitates in EP tubes No. 1 and No. 2 were creamy yellow. This indicates that the purple protein operon successfully replaced the 134.3 kb long fragment and was successfully expressed in strain No. 3. Although the 134.3 kb long fragment was successfully deleted, the operon was not fully inserted into the genomes of strains No. 1 and No. 2.

### 2.7. Metabolic Engineering of B. subtilis for Isobutanol Production

To explore the productive potential of deletion mutants in metabolic engineering, we constructed isobutanol-producing strains and analyzed the titers. In traditional metabolic engineering, introducing a high-copy-number fermentation plasmid is a commonly used strategy to overexpress enzymes related to the target products. First, we used the plasmid pHT01 as a backbone for the expression of the heterologous Ehrlich pathway, which includes two key genes, *kivD* and *adhA*. According to Li et al., AlsS plays a dominant role in the KIV precursor biosynthetic pathway and its expression level affects the titer of isobutanol [28]. Therefore, we constructed plasmid pCas-69 to overexpress the *alsS*, *ilvC* and *ilvD* genes in *B. subtilis* to address their low expression (Figure 7a). Among the five genes, *ilvC* and *ilvD* came from *E. coli*, *alsS* came from *B. subtilis* [66], and *kivD* and *adhA* came from *Lactococcus lactis* [19]. In *B. subtilis*, the intermediate product pyruvate produced by the glycolysis of glucose is converted by acetolactate synthase (AlsS) to 2-acetolactate, which is then converted by ketoacid reductoisomerase (IlvC) to 2,3-dihydroxy-isovalerate, followed by dihydroxy acid dehydrogenase (IlvD) to 2-ketoisovalerate decarboxylase (KivD) and to isobutyraldehyde. In the presence of ethanol dehydrogenase (AdhA), isobutanol can finally be synthesized (Figure 7b).

The wild-type strain GI0 and ten deletion mutants (GI1-3, G5-8 and G10-12) were subjected to microaerobic fermentation in shake flasks containing 20 mL of LBGSM-I medium. Three parallel sets of experiments were set up with the same seed solution for each strain. During fermentation, samples were taken every 24 h to measure isobutanol, 2/3 MB and acetoin titers. As a result, five deletion mutants (GI1, GI5, GI10, GI11 and GI12) were found to have significantly different isobutanol titers after 72 h of fermentation compared to the wild-type strain GI0 (Figure 7c). Among them, the strain GI12 reached the maximum isobutanol titer of 201.7 mg/L after 72 h of fermentation, which was 2.4 times that of the strain GI0. In addition, four deletion mutants (GI1, GI3, GI10 and GI12) had significantly different 2/3 MB titers compared to strain GI0. Among them, the 2/3 MB titer of strain GI10 was the highest, which was 326.9 mg/L, 2.1 times higher than that of strain GI0. Overall, the genomically simplified deletion mutants exhibited higher isobutanol and 2/3 MB titer compared to the wild-type strain GI0. This result suggested that some genes affecting the isobutanol titer may be located in these deletion fragments and that their absence allows more intermediates to flow to the isobutanol and 2/3 MB metabolic pathways.

## 3. Discussion

The modification of the *B. subtilis* genome, an ideal host for industrial production, has received much attention, and there is a desire to construct simplified cell factories capable of synthesizing compounds that meet the production requirements [67,68]. The development of CRISPR-Cas9 systems has certainly given a new impetus to chromosome modification. However, few CRISPR-Cas9-assisted methods have been developed for genome engineering in *B. subtilis*. In previous reports, particular systems that enable long fragment editing have only been proposed by Altenbuhner [53] and So et al. [56]. In this study, we developed a CRISPR-Cas9-assisted Mtb-NHEJ system for gene editing in *B. subtilis*. The method uses the NHEJ repair mechanism to repair double-strand breaks without relying on any editing templates. To our knowledge, this is the first successful attempt to use the NHEJ mechanism for gene deletion in *B. subtilis*. Approximately 500 bp in the DNA sequences of the extracellular protease genes *nprB* and *vpr* in *B. subtilis* has been successfully deleted by using this method, with a positivity rate of 25% for each. Originally, this system was constructed to provide a more convenient deletion of long fragments of the genome, but the results were not satisfactory. The reason may be that the Mtb-NHEJ system is inefficient in *B. subtilis* and is not sufficient to repair long fragment deletions in the genome. According to Zheng et al. [65], the NHEJ system from Msm, Mtb and Bs was tested in *E. coli*, and the result indicated that among these systems, the Msm-NHEJ system was the best. Therefore, future attempts could be made to use Msm-NHEJ to assist in the deletion of long fragments in the *B. subtilis* genome. With the establishment of an efficient NHEJ system in *B. subtilis*, each round of editing only needs to introduce DSBs at different target sites, which can further achieve continuous gene inactivation or deletion of long genomic fragments.

Generally, the CRISPR-Cas9 system is combined with heterologous DSB repair systems, including homologous recombination systems or NHEJ systems. NHEJ-mediated methods generate stochastic DNA indels in the target region, which makes genome editing inaccurate. In contrast, homologous recombination-mediated methods can achieve precise genome editing with higher editing efficiency. The method developed in this study is based on the CRISPR-Cas9-assisted recombineering, which requires an artificial donor DNA as the editing template. Both circular DNA (plasmid-borne dsDNA) and linear DNA (PCR-amplified dsDNA) [69] or synthesized ssDNA [70] have been used as the editing template in existing methods. Editing templates integrated into a plasmid can avoid attack by DNA exonucleases and lead to it copying itself along with plasmid replication, which greatly increases homologous recombination frequency and thus increases editing efficiency. The genome-editing method has been proven to be efficient for large fragment editing. It enabled us to delete DNA fragments of up to 134.3 kb and to achieve the integration and expression of exogenous manipulators into the genome. Compared with existing methods, our method results in higher positivity rates and longer deleted fragments. Based on the analysis of the available experimental results, we believe that the position of the DNA fragment in the genome and the selected sgRNA sequence may also influence editing efficiency. In addition, we have improved the process of gene editing (Appendix A). First, the plasmid was successfully transformed, and then single colonies were selected and enrichment culture was performed. D-mannose was then added to the bacteria during growth to induce Cas9 expression. This approach is a good solution to the problem of editing efficiency being compromised because of the low transformation efficiency of *B. subtilis*. Theoretically, fully expressing genes mediating HDR and NHEJ before Cas9 cuts the chromosome can improve the efficiency of recombination and repair.

Homologous recombination between the cleaved genome and the plasmid may produce either a non-crossover product or a crossover product. There is a certain probability that the entire plasmid will be integrated into the genome at the target site. Moreover, all the components of this genome editing system are located on a single plasmid, making the plasmid too large, which can harm transformation efficiency and editing efficiency. Sachla et al. experimented with designing repair templates as PCR products or specific genes and structures in *B. subtilis*, which facilitated genomic manipulation without constructing plasmids [71]. To avoid these problems, and to address the issues associated with plasmid construction, we also attempted to use linear DNA as an editing template for long fragment deletions. As a result, a long fragment of 134.3 kb was deleted, with a positivity rate of 87.0%. This confirms the feasibility of using linear DNA as an editing template in *B. subtilis* and provides an additional option for genetic engineering experiments in *B. subtilis*.

As a powerful genome engineering tool, the method has great potential for application. We applied the method in genome simplification and metabolic engineering. Engineering biological systems to achieve a specific purpose is one of the goals of synthetic biology [72]. To achieve this, rational deletions can be made in the genomes of microorganisms. The resulting simplified cells have predictable behavior and can provide a platform for the construction of various genetic systems [41]. In previous studies, researchers explored nonessential sequences and attempted to remove them individually or cumulatively from the *B. subtilis* genome in an attempt to construct a minimized genome. However, these methods, including selectable marker cassettes, counter-selectable marker systems, specific recombinases, etc., are complicated, time-consuming, and inefficient. In comparison, our approach shows great advantages. In our study, the deletion of 10 long fragments of DNA did not have a significant effect on the growth of the cells. By placing multiple long fragments of the sgRNA expression cassette and the donor DNA template on the same plasmid, we achieved the simultaneous deletion of multiple long fragments. This strategy allowed us to rapidly obtain cells with different combinations of deletion fragments, shortening the editing cycle as a result. In the course of our research, we found it interesting that sometimes using primers outside the homology arm and primers inside the deleted fragment were both able to amplify the corresponding PCR products. However, after being re-streaked on LB agar plates and following plasmid curing, PCR products were only generated by amplification with primers outside the homology arms. The occurrence of this phenomenon may be related to the following conjecture. First, *B. subtilis* colonies are large, with rough and irregular surfaces, so there is a high probability that successfully edited colonies will join and mix bacteria with unedited colonies. Furthermore, we suspect that the bacteria were edited during growth and that the DNA ends on the genome were successfully repaired, but the long fragments cut off by Cas9 were not immediately degraded. As previously reported, DNA-damaged *B. subtilis* causes a series of physiological responses, called the SOS response [73], which inhibits cell division and thus causes the bacteria to appear filamentous [74,75,76]. Filamentous bacteria contain multiple nuclei, and perhaps this is the reason why PCR products can be amplified using both primer pairs.

Nowadays, the continuous demand for microbial platforms with high capacities to produce bioproducts at low costs is rapidly increasing [77]. *B. subtilis* is regarded as one of the ideal cell factories [27]. To explore the application of deletion mutants in metabolic engineering, we used plasmids carrying genes related to isobutanol synthesis to screen overproducers from deletion mutants. The GI12 strain obtained from the screening produced 201.7 mg/L of isobutanol in shake flasks, which was 2.4 times higher than that of the wild-type strain GI0. This is a good indication that the deletion mutant library constructed by this method has great potential for application in metabolic engineering. It is worth mentioning that we found a significant amount of acetoin in the fermentation product, which was produced by the decarboxylation of the intermediate product 2-acetolactate (ACLAC), with a maximum titer of 3.7 g/L after 48 h of fermentation. We observed that after 72 h of fermentation, the titer of isobutanol increased to some extent in different strains, while the content of acetoin decreased accordingly in all of them. It suggests that the poor isobutanol titer is most likely related to the accumulation of acetoin. In subsequent studies, improvements in the enzymatic activity of IlvC and IlvD and the amount of NADPH will probably provide effective ways to further increase isobutanol titer. In addition, the morphology of *B. subtilis* before and after 72 h fermentation was observed under a microscope by Gram staining [78]. Significant morphological changes were observed in all strains after 72 h fermentation due to inadequate nutritional conditions in the environment during the later stages of fermentation (Appendix A). In particular, wild-type strain GI0, and genomic deletion strains GI1 and GI2 formed spores. As indicated in the previous study, Morimoto et al. simplified the genome of *B. subtilis* and found the transition period was prolonged due to the absence of genes essential for spore formation, leading to an increase in extracellular cellulase and protease production [41]. From this observation, we deduced that if the genes associated with spore formation could be inactivated, isobutanol titer might be further increased.

## 4. Materials and Methods

### 4.1. Strains and Cultural Conditions

*E. coli* strain JM109 was used as the host strain for molecular cloning and plasmid construction operations. *B. subtilis* 168 served as the genetic material for the editing experiments. The transformation of *E. coli* and *B. subtilis* was performed by the heat shock method [79] and the two-step transformation procedure [80], respectively. The strains involved in this study are listed in Appendix A. Details of the reagents and media used in this study are listed in Appendix A. Unless stated otherwise, a Luria-Bertani (LB) medium (10 g/L tryptone, 5 g/L yeast extract, and 10 g/L NaCl) was used for cell growth. The solid medium contained 20 g/L agar. Microaerobic fermentation of *B. subtilis* was carried out using an LBGSM-I medium. The LBGSM-I medium was composed of LB medium (10 g/L tryptone, 5 g/L yeast extract, and 5 g/L NaCl) with 20 g/L glucose, 100 mmol/L potassium phosphate buffer (pH = 7.0), and 1000 dilution of Trace Metal Mix A5 [19,28]. For plasmids with pCas-sgRNA-X (X = 1, 2, 3…, 12) as the backbone, kanamycin was used for screening in *E. coli* and *B. subtilis* at final concentrations of 50 μg/mL and 20 μg/mL, respectively, while plasmids with pHT01 as the backbone were screened in *E. coli* using ampicillin at a final concentration of 100 μg/mL and in *B. subtilis* using chloramphenicol at a final concentration of 10 μg/mL. To induce the CRISPR-Cas9 system in *B. subtilis* cells, 0.2% D-mannose was added.

### 4.2. Plasmid Construction

The plasmids used in this study are listed in Appendix A. The detailed construction procedures for editing plasmid pCas-sgRNA-X are shown in Appendix A. The CRISPR target sequences designed in this study are described in Appendix A. The construction of plasmids pCas-sgRNA-X is the basis for genome editing experiments. First, the pJOE8999 [53] plasmid was used as the parental plasmid. Primers were designed between the promoter P_vanP_ and the gRNA scaffold. A 20 bp specific spacer region of the target sequence was added to the primers as an overlap region. The PCR product was transformed directly into JM109 by single PCR. Fragments were self-conjugated into a loop to construct a plasmid containing an sgRNA expression chimera. Specific donor DNA consists of two homologous arms of similar length obtained by PCR from the wild-type *B. subtilis* genome. The donor DNA was inserted into the plasmid containing the single sgRNA by Gibson Assembly. The plasmid containing an sgRNA expression chimera was thus successfully constructed. For the construction of plasmids containing two sgRNA expression chimeras, only the second sgRNA, including P_vanP_, spacer and gRNA scaffold, needs to be inserted into the plasmid by Gibson Assembly.

To construct plasmid pHT01-Mtb-NHEJ, the *ligD* and *mku* genes from *Mycobacterium tuberculosis* H37Rv [62] were inserted downstream of promoter P_grac_ on the pHT01 plasmid via Gibson Assembly. Similarly, to construct plasmid pHT01-65, *kivD* and *adhA* genes from *Lactococcus lactis* [19], were inserted into the pHT01 plasmid via Gibson Assembly. The *ilvC* and *ilvD* genes from *E. coli*, and the *alsS* gene from *Bacillus subtilis* [66] were inserted into plasmid pJOE8999 [53], downstream of promoter P_manP_, replacing the Cas9-encoding gene.

### 4.3. Genome Editing Procedure

Details of the genome editing protocol are provided in Appendix A. First, Plasmid pCas-sgRNA-X carrying the kanamycin resistance gene was transformed into wild-type *B. subtilis* 168, and positive transformants were obtained on LB agar plates containing kanamycin at 30 °C. A single colony was randomly picked and inoculated into a 5 mL LB medium and incubated at 30 °C until OD_600_ = 0.4–0.6. Then, 0.2% of D-mannose was added to the culture and incubation continued for 8 h. A total of 1 μL of culture was taken and serially diluted and spread on LB plates containing kanamycin and D-mannose. Positive mutants were verified by colony PCR and sequencing. The validation primers used in the genome editing experiments are listed in Appendix A.

### 4.4. Calculation of Positivity Rate

Positive mutants were screened by colony PCR on random colonies on LB plates containing kanamycin and D-mannose and the positive mutants were sequenced for further validation. The number of positive clones screened as a proportion of the total number of colonies identified was used to calculate the positivity rate.

### 4.5. Plasmid Curing

Cas9 transcript expression under P_manP_ control was first induced by incubation on LB agar plates containing 20 μg/mL kanamycin and 0.2% D-mannose at 30 °C. The identified positive mutants were inoculated in a resistance-free liquid LB medium and incubated for 12 h at 50 °C with 220 rpm shaking. They were scribed on LB agar plates without kanamycin to obtain single colonies at 42 °C. Finally, single colonies were inoculated into an LB medium with or without corresponding antibiotics for further verification. The complete plasmid curing process is shown in Appendix A.

### 4.6. Measurement of Growth Curve

To determine the growth curve, single colonies were inoculated into a 5 mL LB medium and incubated at 37 °C for 12 h. Then, 200 μL of the seed solution was taken and inoculated into 20 mL of LB medium in a 50 mL conical flask and incubated at 37 °C. Samples were taken every 30 min and the optical density of the culture was measured OD_600_ using a V-5100 UV spectrophotometer (V-5100, Shanghai Metash Instruments Co., Ltd, Shanghai, China).

### 4.7. Shake Flask Fermentation and Isobutanol Detection

Single colonies were inoculated in a 5 mL LB medium containing antibiotics and incubated at 37 °C for 12 h. Then, 200 μL of the culture was inoculated into 20 mL of the LBGSM-I medium containing antibiotics, isopropyl-beta-D-thiogalactopyranoside (IPTG), and D-mannose, and placed in a 250 mL screw-cap conical flask for microaerobic fermentation. Samples were taken every 24 h and the biomass was measured by measuring the OD_600_ of the fermentation broth using a V-5100 UV spectrophotometer (V-5100, Shanghai Metash Instruments Co., Ltd, Shanghai, China). The fermentation broth was centrifuged at 12,000 rpm for 10 min and the supernatant was used to configure the sample. High purity isobutanol was used as the standard, high purity n-pentyl alcohol was used as the internal standard, and the isobutanol potency was measured using a gas chromatograph (PANNA GCA91, Shanghai Wangxu Electric Co., Ltd, Shanghai, China).

## 5. Conclusions

In conclusion, the present study provided an efficient CRISPR-Cas9 genome editing method which facilitated *B. subtilis* genome simplification. The 12 non-essential regions in the *B. subtilis* 168 were deleted separately, ranging from 17.6 to 134.3 kb in length. The simultaneous deletion of two long fragments was also achieved. A deletion mutant strain with a cumulative deletion of 334.0 kb of the chromosome was constructed. Through a reduction in donor DNA, it was demonstrated that the high efficiency of the HR method mediated by CRISPR-Cas9 required at least a 1000 bp homologous repair template. The linear donor DNA as a repair template could leave out the construction of editing plasmids. Isobutanol producers were further constructed by using wild-type strain and deletion mutant strains as chassis cells. In the shake flask fermentation, the 134.3 kb fragment of the deletion mutant (GI12) accumulated a 201.7 mg/L titer of isobutanol, which is 2.4 times that produced by the wild-type strain GI0. Although this isobutanol overproducer could be further optimized, especially in the regulation of isobutanol titer, this efficient method would be a promising tool by which to promote *B. subtilis* as a cell factory for a variety of bioproducts.

## Figures and Tables

**Figure 1 ijms-23-04853-f001:**
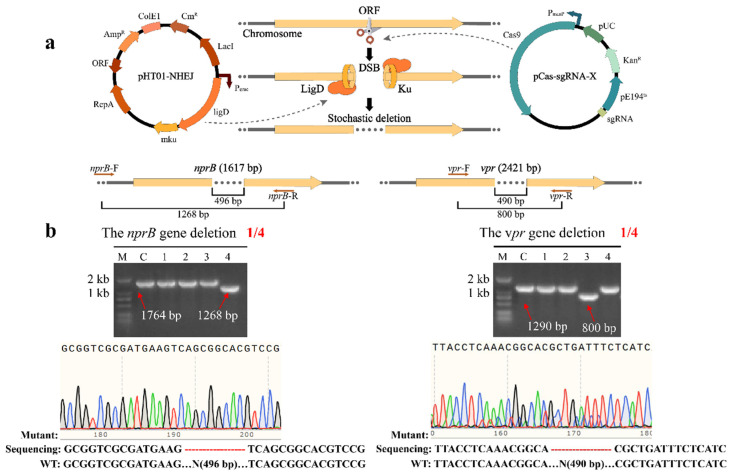
Repair of DNA double-strand breaks by the NHEJ system. (**a**) Deletion of extracellular protease genes in *B. subtilis* using CRISPR-Cas9-assisted non-homologous end-joining editing. (**b**) PCR verification of the *vpr* and *nprB* gene deletions.

**Figure 2 ijms-23-04853-f002:**
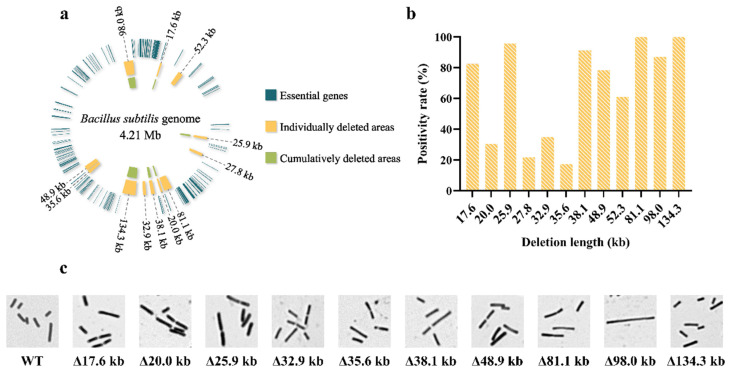
The genomic positions and positivity rates of long fragment deletions (**a**) Position of each deleted long fragment in the genome. (**b**) Positivity rate statistics of each long fragment deletion. (**c**) Morphological comparison of the wild-type strain (WT) and long fragment deletion strains. All strains were Gram-stained and observed using light microscopy.

**Figure 3 ijms-23-04853-f003:**
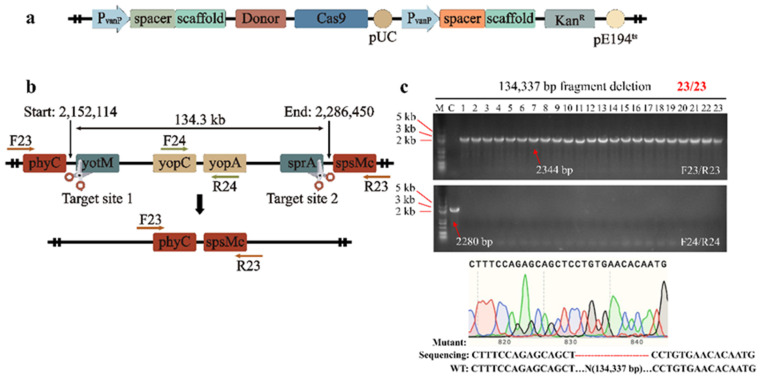
The design of 134.3 kb long fragment deletion (**a**) Construct of plasmid pCas-sgRNA-X series for genomic deletion. Cas9 protein was expressed by a mannose-induced promoter and sgRNAs were expressed by a constitutive promoter. (**b**) Deletion of 134.3 kb genomic fragment in *B. subtilis*. (**c**) Representative PCR results of a 134.3 kb fragment deletion. Colonies were randomly picked for PCR screening, and a wild-type colony served as control (abbreviated to C). Primers F23/R23 and F24/R24 were used to prove the fragment was successfully deleted. If so, the F23/R23 would have a positive PCR result whereas F24/R24 would not obtain any result.

**Figure 4 ijms-23-04853-f004:**
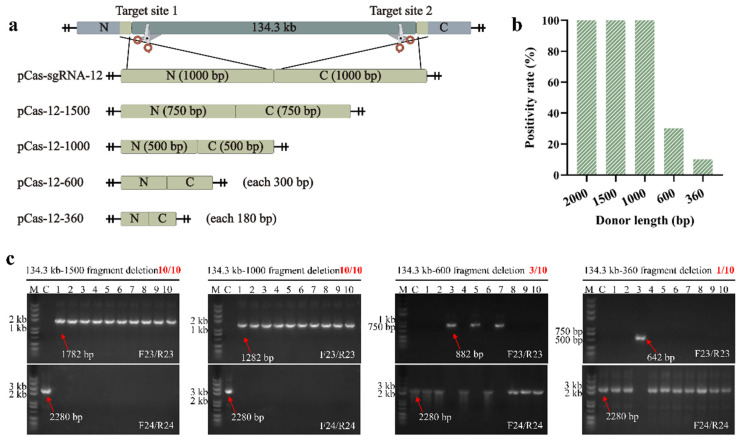
Effect of donor DNA size on the positivity rate of 134.3 kb fragment deletion (**a**) The lengths of the donor DNA templates are 2000, 1500, 1000, 600 bp and 360 bp, respectively. (**b**) The positivity rate of 134.3 kb fragment deletion under different lengths of donor DNA. (**c**) Using primers F23/R23 and F24/R24, the 134.3 kb fragment deletions mediated by different lengths of donor DNA (15,001,000,600 and 360 bp) were verified.

**Figure 5 ijms-23-04853-f005:**
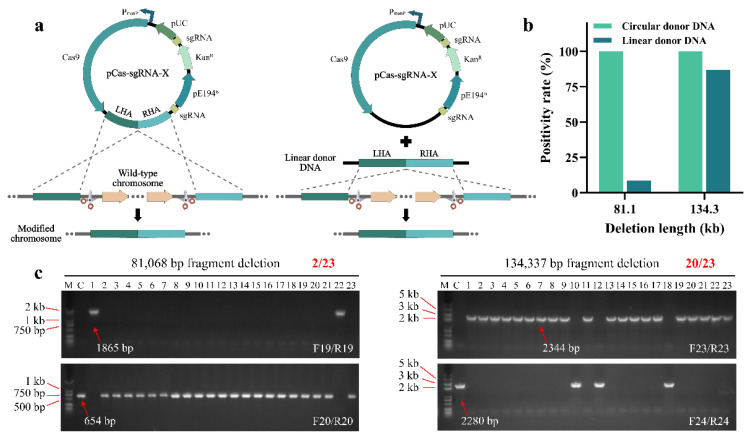
Long fragment deletions mediated by linear donor DNA. (**a**) The different designs of circular and linear donor DNA. (**b**) The positivity rates for 81.1 and 134.3 kb fragment deletions under different forms of donor DNA conditions. For 81.1 and 134.3 kb fragment deletions, the positivity rates mediated by the circular and linear DNA were compared. (**c**) Using linear donor DNA as repair template, PCR results were verified for 81.1 and 134.3 kb fragment deletions.

**Figure 6 ijms-23-04853-f006:**
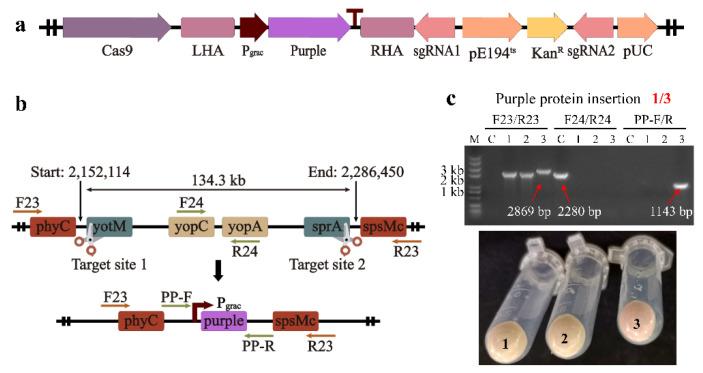
Stable expression of the purple protein in the genome (**a**) The operon P_grac_-purple was inserted between the left homologous arm (LHA) and the right homologous arm (RHA) and integrated into the genome by HDR. (**b**) The purple protein was integrated into the original genome at the location of a 134.3 kb long fragment. (**c**) Purple protein was successfully inserted into the genome of colony No. 3 and expressed, as verified by PCR results and the color of colonies.

**Figure 7 ijms-23-04853-f007:**
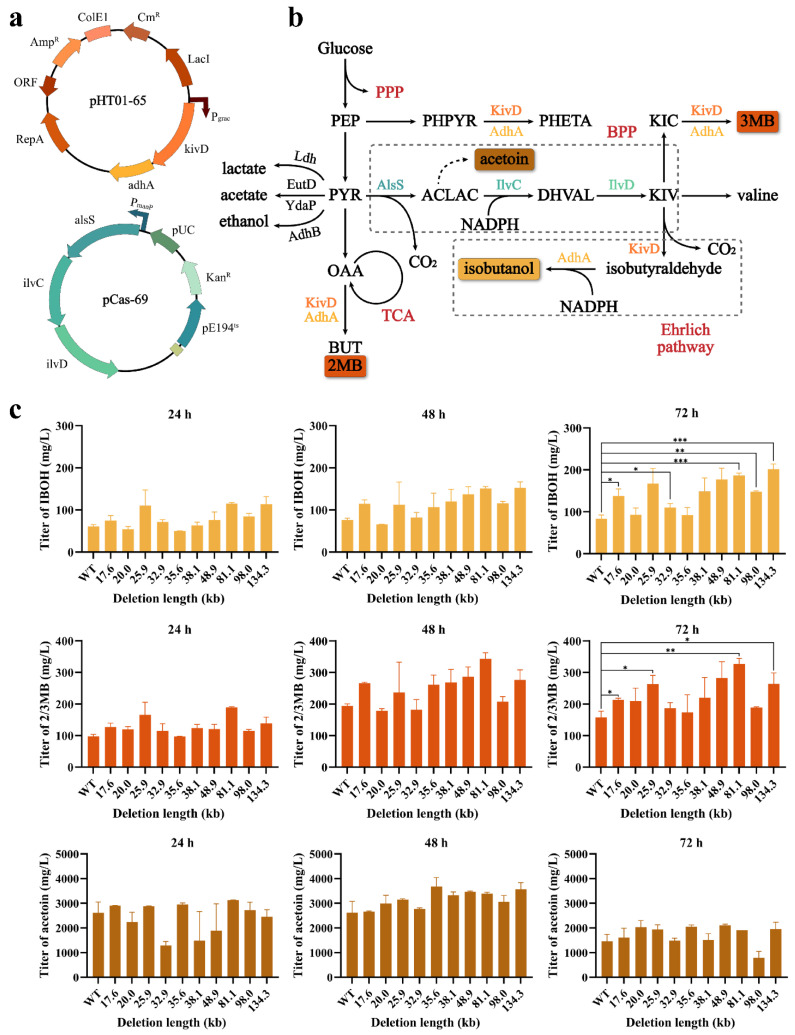
Production pathways for isobutanol and 2/3MB (**a**) The promoter P_grac_ of plasmid pHT01-65 was induced by IPTG to initiate transcription and expression of the *kivD* and *adhA* genes. The promoter P_manP_ of plasmid pCas-69 initiates transcription and expression of the *alsS*, *ilvC* and *ilvD* genes upon mannose induction. (**b**) Schematic representation of heterologous pathway leading to isobutanol and 3MB formation in *B. subtilis*. PEP (phosphoenolpyruvate), PYR (pyruvate), OAA (oxaloacetate), ACLAC (acetolactate), DHVAL (2,3-dihydroxy-3-methyl butanoate), KIV (2-ketoisovalerate), KIC (2-ketoisocaproate), BUT (butanol), IBOH (isobutanol), 2MB (2-methyl-1-butanol), 3MB (3-methyl-1-butanol), PHPYR (phenylpyruvate), PHETA (2-phenyl ethanol), NADPH (nicotinamide adenine dinucleotide phosphate), PPP (pentose phosphate pathway), TCA (citrate cycle), BPP (biosynthetic 2-ketoisovalerate precursor pathway). (**c**) Results of microaerobic fermentation of wild-type strain GI0 and ten deletion mutants (GI1-3, GI5-8 and GI10-12) at 30 °C for 72 h. Orange represents isobutanol titers produced by 11 strains, red represents titers producing a mixture of 2MB and 3MB, and brown indicates acetoin titers. Values and error bars represent mean and s.d. (*n* = 3), respectively. * *p* < 0.1, ** *p* < 0.01, *** *p* < 0.001 as determined by a two-tailed *t*-test.

## Data Availability

Not applicable.

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
