# Peer review of "Efficient Large-Scale and Scarless Genome Engineering Enables the Construction and Screening of Bacillus subtilis Biofuel Overproducers"

_ijms, 2022, doi:10.3390/ijms23094853_

Round 1

Reviewer 1 Report

The science in this manuscript seems fine, and the use of the Mtb NHEJ products to facilitate proper deletion constructs seems like a significant contribution.

However, the manuscript is riddled with incorrect English usage throughout which often makes reading the difficult. Examples in the Abstract alone are: "positive" is used which should be "positivity"; liner DNA should be linear DNA; "serial" should be "series"; should read "difficulty of plasmid construction", and "new generation biofuel." However, there are many, many more examples, and careful editing will be needed to improve the language in the manuscript.

l 36 - might give an example here of the "bottom-up" strategy.

Author Response

Response to Reviewer 1:

Point 1: The science in this manuscript seems fine, and the use of the Mtb NHEJ products to facilitate proper deletion constructs seems like a significant contribution.

However, the manuscript is riddled with incorrect English usage throughout which often makes reading the difficult. Examples in the Abstract alone are: "positive" is used which should be "positivity"; liner DNA should be linear DNA; "serial" should be "series"; should read "difficulty of plasmid construction", and "new generation biofuel." However, there are many, many more examples, and careful editing will be needed to improve the language in the manuscript.

Response 1: Thanks for your corrections. We have revised the English usage in the main text and figures. Other grammatical issues were also corrected.

Using the mode of “Track Changes”, in line 9 “to” was changed to “, which can”;

in line 12 “in Bacillus subtilis genome” was changed to “in the Bacillus subtilis genome”;

in line 14 “a positive rate of” was changed to “a positivity rate of”;

in line 14 “Whether the length of the donor DNA can affect the positive rate was also investigated. To decrease the difficulty for plasmid construction, we further built a CRISPR-Cas9 system with liner donor DNA and a heterologous NHEJ system without the requirement of donor DNA to achieve gene inactivation in Bacillus subtilis, both of which have not been reported before.” was replaced with “The effects of using a heterologous NHEJ system, linear donor DNA, and various donor DNA length on the engineering efficiencies were also investigated.”;

in line 21 “a serial of” was changed to “a series of”;

in line 22 “a new generation of biofuel.” was changed to “a new generation biofuel.”;

in line 30 “Synthetic biology has made rapid development and shown great potential in biosensing, therapeutics and the production of novel biomaterials [1, 2].” was replaced with “Synthetic biology has made rapid development, showing great potential in biosensing, therapeutics and production of novel biomaterials [1, 2].”;

in line 42 “. This strategy” was changed to “, which”;

in line 47 “In order to” was changed to “To”;

in line 48 “withstand process pressures” was changed to “tolerate the stresses in the process”;

in line 49 “Bacillus subtilis (B. subtilis), which is highly tolerant to high concentrations of isobutanol, is regarded as an ideal host for isobutanol production [23, 24]. Consequently, the construction of B. subtilis with high isobutanol production is of great importance in metabolic engineering.” was replaced with “Bacillus subtilis (B. subtilis), a Gram-positive GRAS (generally recognized as safe) bacterium [25-27], which is highly tolerant to high concentrations of isobutanol, is considered to be an excellent industrial host bacterium for the production of bulk chemicals [28, 29] such as isobutanol [23, 24]. Taken together, the construction of B. subtilis with high isobutanol production is of great importance.”;

in lines 57-72 “B. subtilis, a Gram-positive bacterium …The constructed mutant strains were used to produce various drugs, proteins and small-molecule chemicals, etc.” was replaced with “To minimize the risk of disruptive effects of the production of bulk chemicals on the cells themselves and the metabolites [31, 32], the B. subtilis genome could be simplified to eliminate nonessential features, concentrate cellular metabolic activity on the desired products and reduce the cost of industrial production [33, 34]. Therefore, the ideal chassis should contain only the minimum set of functions required for synthetic production [12, 35]. Long fragment editing technology was used to accelerate the modification of bacterial genomics and obtain an ideal chassis to construct ideal hosts for producing various drugs, proteins, and small-molecule chemicals, etc [36].”;

in line 81 “Westers et al. constructed a mutant strain with 7.7% reduction in genome, whose growth activity was unaffected [37].” was replaced with “Westers et al. constructed a mutant strain with a 7.7% reduction in the genome, without changing the growth rate [37].”;

in line 85 “Morimoto et al. deleted the 874.0 kb (20.7%) DNA sequence, while the production of extracellular cellulase and protease was significantly increased [42].” was replaced with “Morimoto et al. sequentially deleted up to 20.7% of DNA sequence (874.0 kb), while the production of extracellular cellulase and protease was significantly increased [42].”;

in line 88 “Li et al. deleted” was changed to “Li et al. sequentially deleted”;

in line 89 “in B. subtilis genome” was changed to “in the B. subtilis genome”;

in line 95 “defend against invasion” was changed to “defends against an invasion”;

in line 100 “has no effect on” was changed to “does not affect”;

in line 100 “Under the guidance of the sgRNA, the Cas9 protein cleaves at the target site where a proper protospacer-adjacent motif (PAM) exists, causing the double-stranded break (DSB) [51].” was replaced with “Under the guidance of the sgRNA, the Cas9 protein cleaves DNA causing the double-stranded break (DSB) [51] at the target site where there is a proper protospacer-adjacent motif (PAM) exists, which is 5’-NGG-3’ for S. pyogenes Cas9 [27].”;

in line 115, 181, 185, 252, 257, 286, 298, 312, 313, 318, 324, 327, 328, 339, 354, 359, 366, 378, 386, 387, 523, and 568 “positive rate” was changed to “positivity rate”;

in line 123 “deletion up to 134.3 kb,” was changed to “deletion of up 134.3 kb,”;

in line 130 “Strains and culture conditions,” was changed to “Strains and cultural conditions”;

in line 138 “using LBGSM-I medium” was changed to “using an LBGSM-I medium”;

in line 150 “is” was changed to “are”;

in line 157, 160, 272 and 281 “an sgRNA” was changed to “a sgRNA”;

in line 157 “which consists of” was changed to “consists of”;

in line 174 “B. subtilis 168 and positive transformants” was changed to “B. subtilis 168, and positive transformants”;

in line 175 “into 5 ml LB” was changed to “into a 5 mL LB”;

in line 195 “into 5 mL of LB” was changed to “into a 5 mL LB”;

in line 215 “It is predicted that there are 271 genes are essential for bacterial growth in LB medium at 37°C, most of which are related to cell membrane structure, genetic information transfer and energy metabolism [63, 65, 66].” was replaced with “According to recent studies, there are 257 genes are essential for bacterial growth in LB medium at 37°C, most of which are related to cell membrane structure, genetic information transfer, and energy metabolism [63, 65, 66].”;

in line 227 “exhibiting” was changed to “exhibits”;

in line 232 “carring” was changed to “carries”;

in line 248 “if there was deletion” was changed to “if there was a deletion”;

in line 249 “As a results” was changed to “As a result”;

in line 251 “deleted in nprB gene” was changed to “deleted in the nprB gene”;

in line 257 “deleted in vpr gene” was changed to “deleted in the vpr gene”;

in line 269 “to homologous recombination” was changed to “to the homologous recombination”;

in line 283 “1880 bp band” was changed to “the 1880 bp band”;

in line 305 “complete deletion” was changed to “the complete deletion”;

in line 329 “wild-type strain” was changed to “the wild-type strain”;

in line 334 “Cas9 protein was expressed by mannose-induced promoter and gRNAs was expressed by constitutive promoter.” was replaced with “Cas9 protein was expressed by a mannose-induced promoter and sgRNAs were expressed by a constitutive promoter.”;

in line 338 “have positive PCR result” was changed to “have a positive PCR result”;

in line 349 “If the 2000” was changed to “If 2000”;

in line 360, 399, 548 “positive rates” was changed to “positivity rates”;

in line 367 “positive rate” was changed to “The positivity rate”;

in line 379 “as repair template” was changed to “as a repair template”;

in line 380 “in order to reduce” was changed to “to reduce”;

in line 409 “were integrated together into” was changed to “were integrated into”;

in line 419 “well be” was changed to “will be”;

in line 433 “inhibit” was changed to “inhibiting”;

in line 434 “elongate” was changed to “elongating”;

in line 456 “the precipitate” was changed to “the precipitates”;

in line 458 “in the strain No. 3” was changed to “in strain No. 3”;

in line 463 “purple protein” was changed to “the purple protein”;

in line 463 “left homologous arm” was changed to “the left homologous arm”;

in line 464 “right homologous arm” was changed to “the right homologous arm”;

in line 469 “In order to” was changed to “To”;

in line 470 “analysed” was changed to “analyzed”;

in line 524 “enable more convenient” was changed to “enable a more convenient”;

in line 528 “result indicated” was changed to “the result indicated”;

in line 528 “Msm-NHEJ system” was changed to “the Msm-NHEJ system”;

in line 562 “have a negative impact on” was changed to “harm”;

in line 566 “we attempted” was changed to “we also attempted”;

in line 606 “the wild type strain” was changed to “the wild-type strain”;

in line 614 “the improvements of the enzymatic” was changed to “the improvements in the enzymatic”;

in line 620 “, genomic deletion strains” was changed to “, and genomic deletion strains”;

in line 643 “acknowledge financial support” was changed to “acknowledge the financial support”.

l 36 - might give an example here of the "bottom-up" strategy.

Response 2: Thanks for your suggestion. The “bottom-up” strategy was added in line 31 “The “bottom-up” approach requires that the synthesized pathways be assembled into long fragments from scratch and that all essential components be ligated into the chromosome [7, 11-14]. However, “bottom-up” approaches are difficult to maintain genomic stability because of the complexity of metabolic processes and interactions in the organism [12, 15]” in the revised manuscript.

Reviewer 2 Report

Accepted after minor revision, 
1-Please include some of the latest research findings and updated reviews during 2021-2022 needed in the introduction and discussion parts. 
2-Please add a conclusion in the of discussion. The conclusion you have provided is quite brief and you must provide sufficient feedback on the main objectives of your study.

Author Response

Response to Reviewer 2:

Point 1: Accepted after minor revision,

1-Please include some of the latest research findings and updated reviews during 2021-2022 needed in the introduction and discussion parts.

Response 1: Thanks for your suggestions. We have included some of the latest research findings during 2021 to 2022

in line 11 “The “bottom-up” approach requires that the synthesized pathways be assembled into long fragments from scratch and that all essential components be ligated into the chromosome [7, 11-14]. However, “bottom-up” approaches are difficult to maintain genomic stability because of the complexity of metabolic processes and interactions in the organism [12, 15]. The “top-down” approach refers to building a cell factory by deleting nonessential genes and metabolic pathways from the natural cell [10, 16-18], which is a more pragmatic approach than “bottom-up” for building an ideal chassis [7, 15].” in the revised manuscript.

in line 39 “Currently, global environmental and resource issues need to be addressed urgently. Higher alcohols such as isobutanol, n-butanol and 3-methyl-1-butanol (3MB), etc, are seen as an ideal alternative to gasoline due to their high energy density and low hygroscopicity [19-21].” in the revised manuscript.

in line 43 “Bacillus subtilis (B. subtilis), a Gram-positive GRAS (generally recognized as safe) bacterium [23-25], which is highly tolerant to high concentrations of isobutanol, is considered to be an excellent industrial host bacterium for the production of bulk chemicals [26, 27] such as isobutanol [28, 29].” in the revised manuscript.

in line 75 “Under the guidance of the sgRNA, the Cas9 protein cleaves DNA causing the double-stranded break (DSB) [50] at the target site where there is a proper protospacer-adjacent motif (PAM) exists, which is 5’-NGG-3’ for S. pyogenes Cas9 [25].” in the revised manuscript.

in line 78 “The DSB can be repaired by error-prone nonhomologous end-joining (NHEJ) or precise homology-directed repair (HDR) [49, 51]. In B. subtilis, there is a natural NHEJ system, which is active mainly in the late growth and spore formation phase [25, 52].” in the revised manuscript.

in line 81 “Moreover, B. subtilis does not require the assistance of a heterologous homologous recombination system, and the exogenous DNA becomes single-stranded and efficiently recombines with homologous chromosomes as it enters the cell [53, 54].” in the revised manuscript.

in line 83 “The CRISPR-Cas9 gene-editing technology breaks through the limitations of traditional methods, improves editing efficiency, and accelerates genome simplifying [55].” in the revised manuscript.

in line 89 “Song et al. used the CRISPR-Cas9 system to integrate a 2.5 kb expression cassette into the B. subtilis genome [57].” in the revised manuscript.

in line 527 “Sachla et al. experimented with designing repair templates as PCR products or specific genes and structures in B. subtilis, which allowed to facilitate genomic manipulation without constructing plasmids [73].” in the revised manuscript.

in line 564 “Nowadays, a continuous demand for microbial platforms with high capacities to produce bioproducts at low costs is rapidly increasing [79].” in the revised manuscript.

in line 565 “B. subtilis is regarded as one of the ideal cell factories [27].” in the revised manuscript.

Point 2: 2-Please add a conclusion in the of discussion. The conclusion you have provided is quite brief and you must provide sufficient feedback on the main objectives of your study.

Response 2: Thank you, we have added a conclusion “In conclusion, the present study provided an efficient CRISPR-Cas9 genome editing method, which facilitated B. subtilis genome simplification. The 12 non-essential regions in the B. subtilis 168 were deleted separately, ranging from 17.6 to 134.3 kb in length. Simultaneous deletion of two long fragments was also achieved. A deletion mutant strain with a cumulative deletion of 334.0 kb of the chromosome had been constructed. Through the reduction of donor DNA, it was demonstrated that the high efficiency of the HR method mediated by CRISPR-Cas9 required at least a 1000 bp homologous repair template. The linear donor DNA as a repair template could leave out the construction of editing plasmids. Isobutanol producers were further constructed by using wild-type strain and deletion mutant strains as chassis cells. In the shake flask fermentation, the 134.3 kb fragment of the deletion mutant (GI12) accumulated 201.7 mg/L titer of isobutanol, which is 2.4 times that produced by the wild-type strain GI0. Although this isobutanol overproducer could be further optimized, especially in the regulation of isobutanol production, this efficient method would be a promising tool to promote B. subtilis as a cell factory for a variety of bioproducts.” in the last part, Section 5 “Conclusion” to conclude the main work of our study.
